# RNA polymerase II pausing factor NELF in CD8+ T cells promotes antitumor immunity

Bogang Wu [1,6], Xiaowen Zhang[1,6], Huai-Chin Chiang[1], Haihui Pan[1], Bin Yuan[1], Payal Mitra[2], Leilei Qi[2], Hayk Simonyan[3], Colin N. Young[3], Eric Yvon[4], Yanfen Hu[2], Nu Zhang [5] & Rong Li [1✉]

T cell factor 1 (TCF1) is required for memory and stem-like CD8+ T cell functions. How TCF1 partners with other transcription factors to regulate transcription remains unclear. Here we show that negative elongation factor (NELF), an RNA polymerase II (Pol II) pausing factor, cooperates with TCF1 in T cell responses to cancer. Deletion of mouse *Nelfb*, which encodes the NELFB subunit, in mature T lymphocytes impairs immune responses to both primary tumor challenge and tumor antigen-mediated vaccination. *Nelfb* deletion causes more exhausted and reduced memory T cell populations, whereas its ectopic expression boosts antitumor immunity and efficacy of chimeric antigen receptor T-cell immunotherapy. Mechanistically, NELF is associated with TCF1 and recruited preferentially to the enhancers and promoters of TCF1 target genes. *Nelfb* ablation reduces Pol II pausing and chromatin accessibility at these TCF1-associated loci. Our findings thus suggest an important and rate-limiting function of NELF in anti-tumor immunity.

[1] Department of Biochemistry & Molecular Medicine, The George Washington University, Washington, DC 20037, USA. [2] Department of Anatomy & Cell Biology, The George Washington University, Washington, DC 20037, USA. [3] Department of Pharmacology & Physiology, The George Washington University, Washington, DC 20037, USA. [4] Department of Medicine, The George Washington University Cancer Center School of Medicine & Health Sciences, The George Washington University, Washington, DC 20037, USA. [5] Department of Microbiology, Immunology & Molecular Genetics, University of Texas Health San Antonio, San Antonio, TX 78229, USA. [6]These authors contributed equally: Bogang Wu, Xiaowen Zhang. ✉email: rli69@gwu.edu

T cells undergo rapid proliferation and acquire effector function after encountering antigens, co-stimulation signals, and inflammatory cytokines[1–3]. Short-lived effector cells often undergo apoptosis after the clearance of foreign antigens. A small subset of effector cells develops into long-lived memory T cells[2]. Both effector and memory T cells play critical roles in controlling virus infection and tumor outgrowth. However, unlike acute virus infection, tumor antigens often persist in an excessive amount and thus induce an exhaustion phenotype of tumor-reactive T cells[4]. Exhausted T cells, which exhibit upregulation of multiple inhibitory receptors and loss of polyfunctionality, have been the major target for reinvigorating T cell function in anticancer immunotherapy[3]. Despite clinical success in adoptive cell therapy and immune checkpoint blockade-based therapeutics[5,6], most patients still cannot benefit from these immunotherapies, partly due to the lack of persistent tumor-reactive memory T cells.

Recent studies reported a number of key transcription factors that control memory T cell differentiation, including TCF1[7], FOXO1[8], MYB[9], BACH2[10], and BATF3[11]. In particular, TCF1 protein, encoded by the gene *TCF7*, plays a critical role in the regulation of T cell development[12], differentiation[13], and effector function preservation during exhaustion[14,15]. TCF1 promotes chromatin accessibility that favors memory T cell differentiation[16]. Meanwhile, TCF1 depletion exacerbates, and enforced TCF1 expression ameliorates T cell exhaustion[15,16]. TCF1 also promotes long-term T cell survival by promoting the anti-apoptotic factor BCL2[15] while suppressing the pro-apoptotic factor BIM[16]. A TCF1+ T cell population is critical in response to immune checkpoint blockade immunotherapy[17]. Although the role of TCF1 in the determination of T cell fate is well established, little is known about the partners that facilitate its function in T cell differentiation-related transcriptional programming.

Pol II pausing plays a pivotal role in regulating metazoan gene expression[18–20]. At the molecular level, promoter-proximal Pol II pausing prevents nucleosome assembly and thus maintains a permissive chromatin structure for future rounds of transcriptional activation in response to environmental stimuli[21]. Recent studies also showed that Pol II pausing and release play a role in transcriptional enhancers[22]. NELFB is one of the four subunits of the NELF complex that controls Pol II pausing[23]. Mouse genetic studies demonstrate that NELFB is indispensable for early embryogenesis[24] and mammary gland development[25]. In addition, NELF is involved in various aspects of adult tissue homeostasis including energy metabolism in the myocardium[26], inflammatory responses in macrophages[27], myofiber repair after injury[28], and maintenance of junctional integrity[29]. However, the functional significance of NELF-dependent Pol II pausing in the context of T cell function remains unclear.

Here, we define a CD8+ T cell-intrinsic role for NELF during antitumor immune response. Using tissue-specific mouse genetic models, we demonstrate that NELFB in CD8+ T lymphocytes is important for antitumor immunity. Mechanistically, NELF is associated with TCF1 and facilitates chromatin accessibility at TCF1-bound transcriptional enhancers and promoters. We further establish that ectopic NELFB expression boosts host antitumor immunity in mouse models and the efficacy of CAR-T immunotherapy.

## Results

**Nelfb is required for antitumor CD8+ T cell function.** To investigate the role of NELF in mature T cell function, we deleted *Nelfb* in a mature T cell-specific manner by crossing *Nelfb^f/f* and distal *Lck-Cre* (*dLck-Cre*) mouse strains. Unlike the proximal *Lck-Cre* system, *dLck-Cre* is only activated after thymocyte positive selection

and therefore has minimal impact on thymocyte development[30,31]. NELFB protein levels were significantly reduced in primary CD8+ T cells, but not in non-CD8+ cells, from *Nelfb^f/f,dLck-Cre* knockout (hereafter KO) mice versus their *Nelfb^f/f* counterparts (Supplementary Fig. 1a, b). Levels of the other three NELF subunits were also decreased in KO mice versus the *Nelfb^f/f* control (Supplementary Fig. 1b), consistent with the previous finding that protein stability of the four NELF subunits is mutually dependent[32]. KO mice had a slightly lower percentage of CD8+ and increased CD4+ T cells than *Nelfb^f/f* controls (Supplementary Fig. 1c). Cell numbers of CD4+ subpopulations, including naïve, central memory, effector memory, and regulatory T (Treg) cells, were comparable between *Nelfb^f/f* and KO (Supplementary Fig. 1d, e). Among CD8+ T cells, the number of naïve T cells was comparable between *Nelfb^f/f* and KO mice (Supplementary Fig. 1f). However, central memory and effector memory T cell populations were significantly reduced in KO mice compared to those in *Nelfb^f/f* mice (Supplementary Fig. 1f), which could be indicative of a defective response of these mice to environmental antigens.

To investigate the role of NELF in T cell response to tumor growth, we challenged *Nelfb^f/f* and KO mice with syngeneic mouse mammary tumor cells (E0771 and AT-3). Tumors grew more robustly in KO versus *Nelfb^f/f* hosts (Fig. 1a–d). Tumor-infiltrating lymphocytes (TILs) contained substantially fewer total leukocytes (Fig. 1e) and CD8+ T cells (Fig. 1f) of KO versus *Nelfb^f/f* tumor-bearing mice. In addition, KO mice had smaller effector memory (Fig. 1g) and proliferative CD8+ T cell populations (Fig. 1h). To ascertain intrinsic defects of CD8+ T cells from KO mice, we adoptively transferred CD8+ T cells from *Nelfb^f/f* or KO mice to immunodeficient *Rag1^−/−* recipient mice and subsequently challenged the chimeric mice with E0771 tumors. As expected, mice that received *Nelfb^f/f* CD8+ T cells exhibited a strong antitumor response compared to those that received vehicles (Fig. 1i, j). In contrast, adoptively transferred KO CD8+ T cells did not confer any tumor-inhibiting activity versus controls (Fig. 1i, j). Tumors harvested from chimeric mice with KO CD8+ T cells had significantly smaller total and effector memory CD8+ populations versus those with *Nelfb^f/f* CD8+ T cells (Fig. 1k, l). These findings demonstrate a CD8+ T cell-intrinsic function of NELF in antitumor immunity.

**Nelfb deletion impairs memory T cell recall response.** Memory T cell response is critical for mediating the protective role of vaccination[33,34]. To determine the impact of *Nelfb* KO on memory function during tumor antigen-initiated vaccination, we chose two aggressive, ovalbumin (OVA)-expressing tumor models: lymphoma E.G7-OVA and melanoma B16-OVA (Fig. 2a and Supplementary Fig. 2a). In both tumor models, non-vaccinated (non-vac), tumor-bearing KO mice tended to have worse survival than their *Nelfb^f/f* counterparts, though the difference was not statistically significant (Fig. 2b and Supplementary Fig. 2b). This is likely because the robust tumor growth outstripped host antitumor immunity. When mice were vaccinated with OVA protein and then challenged with tumor cells, vaccination significantly lengthened survival in both *Nelfb^f/f* and KO hosts. However, vaccination exhibited substantially greater protection against tumors in *Nelfb^f/f* than KO mice ($p = 0.002$ in Fig. 2b, $p = 0.01$ in Supplementary Fig. 2b). In addition, vaccination markedly increased CD8+ T cell abundance in *Nelfb^f/f*, but not KO, hosts (Fig. 2c), suggesting that *Nelfb* KO impairs memory T cell response.

To ascertain the role of NELF in memory T cell function, we next used heat-inactivated B16 tumor cells as the vaccine based on an established protocol[35]. Consistent with the findings from OVA-mediated vaccination, B16 tumor growth was comparable

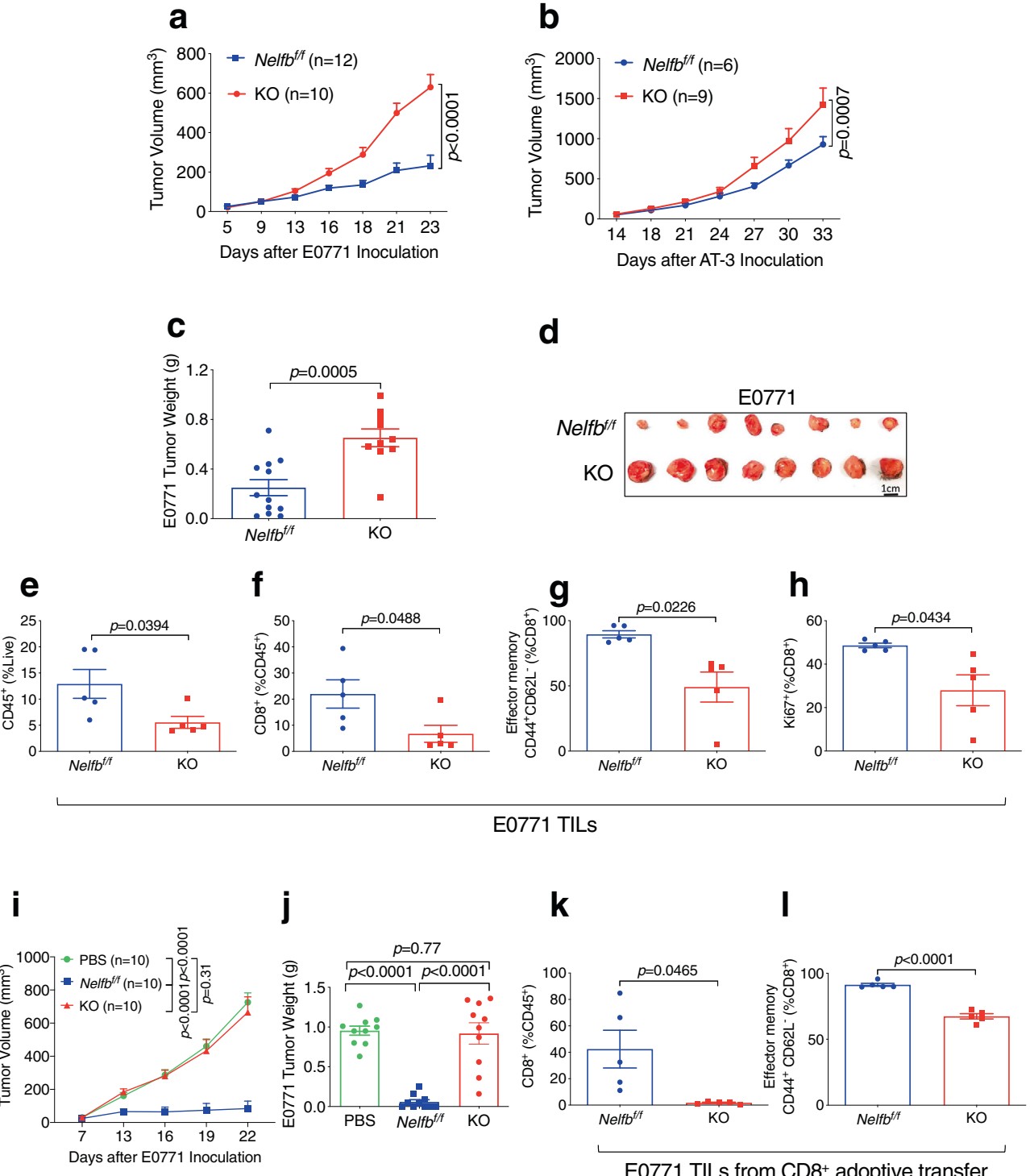

**Fig. 1 *Nelfb* is required for CD8$^+$ T cell-intrinsic antitumor function. a**, **b** Growth curves of E0771 (**a**) and AT-3 (**b**) tumors in *Nelfb$^{f/f}$* and KO mice. **c**, **d** E0771 tumor weight (**c**) and image (**d**) upon harvest in *Nelfb$^{f/f}$* and KO mice. **e–h** TIL analysis for E0771 tumors in *Nelfb$^{f/f}$* and KO mice: **e** CD45$^+$ (% of all live cells), **f** CD8$^+$ (% of CD45$^+$), **g** effector memory CD44$^+$CD62L$^-$ (% of CD8$^+$), **h** Ki67$^+$ (% of CD8$^+$), *n* = 5/group. **i**, **j** E0771 tumor growth curve (**i**) and weight (**j**) in *Rag1$^{-/-}$* mice receiving *Nelfb$^{f/f}$* or KO CD8$^+$ T cells. **k**, **l** TIL analysis for E0771 tumors in *Rag1$^{-/-}$* mice receiving *Nelfb$^{f/f}$* or KO CD8$^+$ T cells. **k** CD8$^+$ (% of CD45$^+$), **l** effector memory CD44$^+$CD62L$^-$ (% of CD8$^+$); *n* = 5/group. Data were presented as mean ± SEM; Student's *t*-test (for two groups) or one-way-ANOVA (for three or more groups). Tumor curves were compared using two-way ANOVA followed by multiple comparisons. Two-sided tests were used. Source data are provided as a Source Data file.

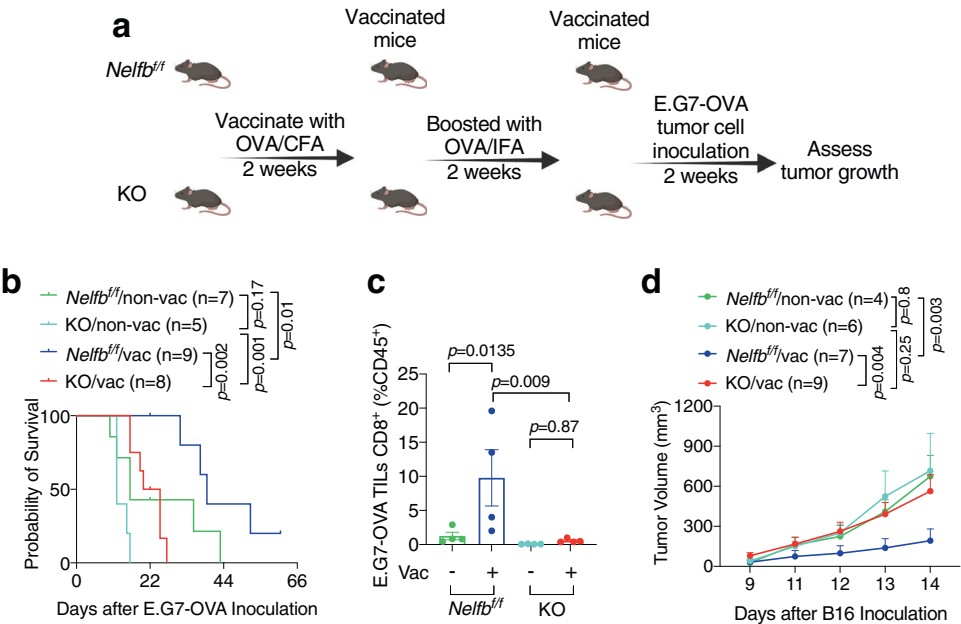

**Fig. 2 *Nelfb* deletion in T cells impairs memory response to tumor antigen vaccination. a** Scheme of vaccination procedure using OVA protein followed by E.G7-OVA tumor challenge. **b** Survival curve for E.G7-OVA tumor-bearing *Nelfb^{f/f}* and KO mice with or without vaccination. **c** CD8⁺ percentages among E.G7-OVA tumor-infiltrating CD45⁺ cells; *n* = 4/group. **d** B16 tumor growth curve in *Nelfb^{f/f}* and KO mice with or without vaccination; *Nelfb^{f/f}* non-vac (*n* = 4), *Nelfb^{f/f}* vac (*n* = 7), KO non-vac (*n* = 6), KO vac (*n* = 9); one-way-ANOVA for comparing mean differences. Tumor curves were compared using two-way ANOVA followed by multiple comparisons. Source data are provided as a Source Data file.

in non-vaccinated *Nelfb^{f/f}* and KO mice (Fig. 2d). In contrast, vaccination provided robust survival benefits in *Nelfb^{f/f}*, but not in KO, mice (Fig. 2d), further corroborating the importance of NELF in memory T cell response. Following vaccination, tumor-bearing *Nelfb^{f/f}* hosts, but not KO ones, displayed significant increases in CD8⁺ T cell abundance in the spleen, and central memory T cells in both spleen and lymph nodes (Supplementary Fig. 2c–e). Taken together, our studies from multiple models of tumor antigen-initiated vaccination demonstrate that NELF plays a pivotal role in promoting tumor antigen-initiated T cell recall response.

***Nelfb* KO causes functional defects in CD8⁺ T cells.** To further characterize CD8⁺ T cell defects in *Nelfb* KO mice, we sought to define cellular functions affected by *Nelfb* loss during T cell receptor (TCR) activation. Carboxyfluorescein succinimidyl ester (CFSE) labeling showed that KO CD8⁺ T cells had relatively normal proliferation rates shortly after in vitro TCR activation (days 1–4), but exhibited significant proliferation defects on day 5 (Fig. 3a and Supplementary Fig. 3a). This indicates that *Nelfb* deletion unlikely directly impaired TCR-activated cell cycle entry. On the other hand, significantly more KO CD8⁺ T cells underwent apoptosis compared to their *Nelfb^{f/f}* counterparts as early as 24 hr after TCR activation (Fig. 3b and Supplementary Fig. 3b).

Interleukin 2 (IL2) is critical for the expansion of activated T cells but also promotes an exhaustion phenotype[36]. As expected, under prolonged in vitro incubation with IL2, both *Nelfb^{f/f}* and KO CD8⁺ T cells gradually increased expression of the immune inhibitory receptors PD1 and TIM3 (Fig. 3c–f). Notably, the magnitude of increase in either PD1/TIM3 single or double-positive exhausted populations was significantly higher in KO versus *Nelfb^{f/f}* cells (Fig. 3c–f). Loss of polyfunctionality is a hallmark of reduced memory stem cell population and increased T cell exhaustion[37]. Extended in vitro proliferation also resulted in a substantially lower abundance of interferon γ (IFNγ) and tumor necrosis factor α (TNFα) double-positive, polyfunctional KO

T cells versus *Nelfb^{f/f}* (Fig. 3g, h and Supplementary Fig. 3c, d). In aggregate, our data strongly indicate that *Nelfb* ablation exacerbates CD8⁺ T cell exhaustion during ex vivo expansion.

To confirm the impact of *Nelfb* deletion on CD8⁺ T cell functionality, we performed single-cell RNA-sequencing (scRNA-seq) using total CD8⁺ T cells isolated from *Nelfb^{f/f}* and KO mice. Based on canonical marker gene expression patterns, we identified four major subsets—naïve, memory, exhausted, and senescent populations (Fig. 4a and Supplementary Fig. 4a). Compared to *Nelfb^{f/f}* CD8⁺ T cells, KO samples had smaller naïve and memory, but larger exhausted and senescent, cell populations (Fig. 4b and Supplementary Fig. 4b–d). Using an established algorithm for trajectory analysis[38], we found two differentiation trajectories initiated from the naïve stage; one leads to the exhausted stage and the other to the senescent stage (Fig. 4c). Pseudotime is a parameter that describes the positioning of individual cells from the differentiation starting point along the specific trajectory, therefore representing the degree of the differentiation from the naïve to terminally differentiated stage[39]. Pseudotime analysis of the differentiation trajectory clearly showed that for both exhausted and senescent lineages, KO cells had significantly higher pseudotimes versus *Nelfb^{f/f}* (Fig. 4d). This indicates that KO cells are further downstream from their differentiation origin and therefore are more terminally differentiated than *Nelfb^{f/f}* cells. Together, our data support the notion that NELF prevents precocious terminal differentiation of CD8⁺ T cells.

**Preferential NELF-dependent Pol II pausing at TCF1 targets.** To elucidate the molecular mechanism by which NELF regulates T cell functionality, we performed NELFB ChIP-seq in primary mouse CD8⁺ T cells. An unbiased analysis of transcription factor binding motif enrichment showed that NELFB chromatin binding overlapped most significantly with that of TCF1 (Fig. 5a)[40]. Of note, NELFB ChIP-seq signals were substantially higher at both enhancers and transcription start sites (TSS) of TCF1

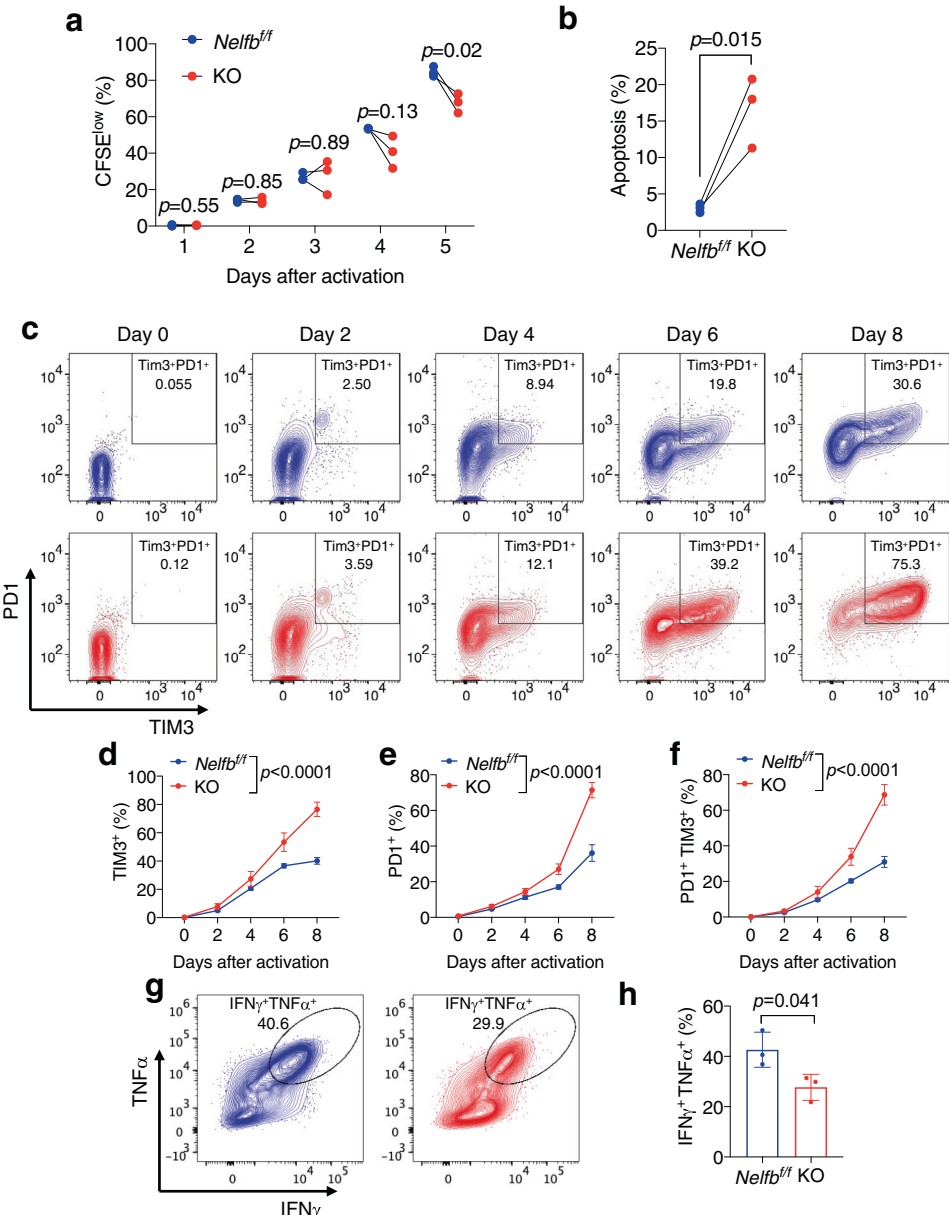

**Fig. 3 *Nelfb* deletion promotes apoptosis and exhaustion of CD8+ T cells. a** Time course of CFSE[low] population as a percentage of *Nelfb*[f/f] and KO CD8+ T cells cultured in vitro, $n = 3$/group. **b** Percentages of apoptotic cells among *Nelfb*[f/f] and KO CD8+ T cells after 24 h of anti-CD3/CD28 activation, assessed by annexin V-APC/PI for flow cytometric analyses, $n = 3$/group. **c** Representative flow cytometry plot of the time course for TIM3+PD1+ double-positive exhausted cells among *Nelfb*[f/f] and KO CD8+ T cells cultured in vitro. **d–f** Percentages of TIM3+ (**d**), PD1+ (**e**), and PD1+TIM3+ (**f**) cells within *Nelfb*[f/f] and KO CD8+ T cells cultured in vitro, $n = 3$/group. **g** Flow cytometry plot of TNFα+IFNγ+ double-positive polyfunctional T cells. **h** Percentages of *Nelfb*[f/f] and KO CD8+ T cells after 10 days of in vitro culture, $n = 3$/group. Data were presented as mean ± SD; mean differences were compared using Student's *t*-test. Time-dependent curves were compared using two-way ANOVA followed by multiple comparisons. Two-sided tests were used. Source data are provided as a Source Data file.

targets[13,41] versus non-TCF1 targets (Fig. 5b). Next, we conducted Pol II ChIP-seq in primary *Nelfb*[f/f] and KO CD8+ T cells. We used Pol II pausing index to assess the degree of promoter-proximal enrichment of Pol II, and value D to denote the maximum difference in vertical distance between two cumulative distributions[42,43] (Fig. 5c). In *Nelfb*[f/f] CD8+ T cells, TCF1 targets had a significantly higher Pol II pausing index versus non-TCF1 targets (Fig. 5c, D = 0.45736, $p < 2.2e{-}16$). *Nelfb* KO decreased the Pol II pausing index to a greater extent at TCF1 targets (D = 0.3172, $p < 2.2e{-}16$) than at non-TCF1 targets (D = 0.13387, $p < 2.2e{-}16$, Fig. 5c, Supplementary Fig. 5a). When enhancers and TSS were analyzed separately, the reduction in Pol II binding

upon *Nelfb* KO was more pronounced at TCF1 targets than at TCF1 non-bound targets (Fig. 5d and Supplementary Fig. 5b). Together, our data indicate that NELF-dependent Pol II binding in primary CD8+ T cells is preferentially associated with TCF1-bound regulatory regions.

**NELF regulates chromatin accessibility of TCF1 targets.** Pol II accumulation at promoters and enhancers have been implicated in chromatin accessibility[21,44]. We, therefore, used an assay for transposase-accessible chromatin sequencing (ATAC-seq) to assess global chromatin openness in primary naive *Nelfb*[f/f] and

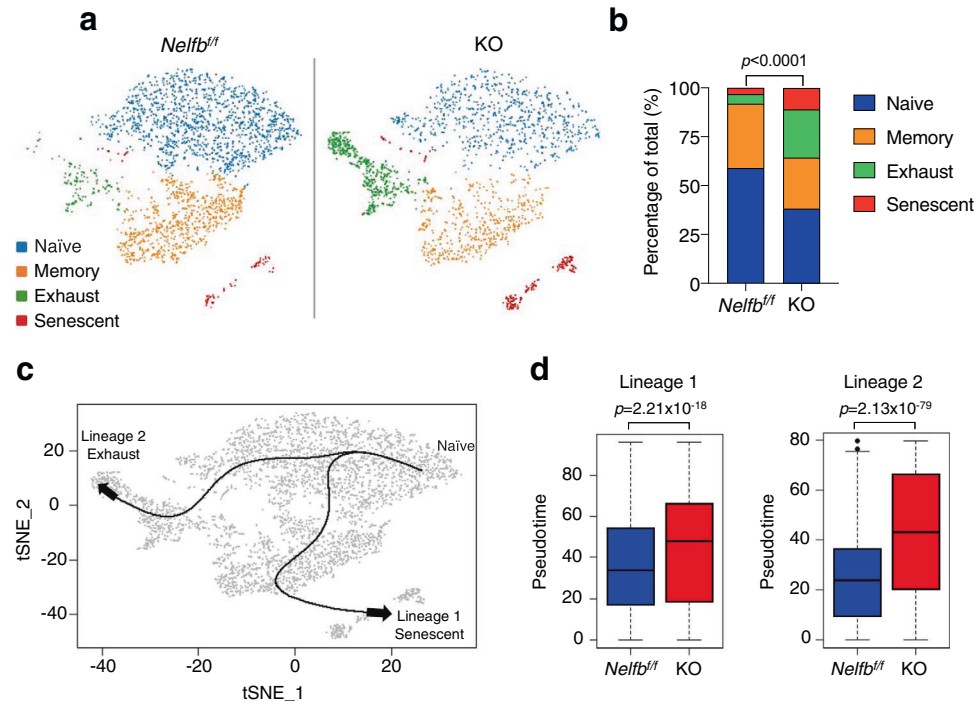

**Fig. 4 *Nelfb* deletion promotes terminal differentiation of CD8$^+$ T cells. a** tSNE plot representing *Nelfb$^{f/f}$* and KO CD8$^+$ T cells isolated from mouse splenocytes at 10-month age. One mouse/genotype. **b** Percentages of individual subpopulations from *Nelfb$^{f/f}$* and KO CD8$^+$ T cells; *Nelfb$^{f/f}$* (n = 2412), KO (n = 1877); frequencies were assessed by Chi-square test. **c** Two differentiation lineages are defined by trajectory analysis. **d** Pseudotime quantification of two differentiation lineages of *Nelfb$^{f/f}$* and KO CD8$^+$ T cells, *Nelfb$^{f/f}$* (n = 2412), KO (n = 1877); two-sided Student's *t*-test. The bounds of the box represent the 25th and 75th percentiles of the interquartile range. The black line in the box interior represents the median of the data. The whiskers represent the minimum and maximum values of the data and the black dot outside the box and the whiskers represent an outlier. Source data are provided as a Source Data file.

KO CD8$^+$ T cells. Principal component analysis showed that *Nelfb$^{f/f}$* and KO had distinct chromatin opening states (Supplementary Fig. 5c). Three-quarters of the differentially accessible regions exhibited reduced chromatin openness upon *Nelfb* deletion (Supplementary Fig. 5d), suggesting a role of NELF in promoting chromatin accessibility. Reminiscent of its enrichment in NELFB chromatin binding regions (Fig. 5a), TCF1 binding was the most significantly enriched motif in the chromatin regions with KO-impaired accessibility (Fig. 6a). *Nelfb* KO-triggered reduction in chromatin openness occurred more at TCF1 targets than non-TCF1 targets, and furthermore, the KO effect was more pronounced at enhancers than TSS of TCF1 targets (Fig. 6b and Supplementary Figs. 5e, 6a, b). These data support the notion that NELF in naïve CD8$^+$ T cells preferentially facilitates chromatin accessibility at TSS and to a greater extent, enhancers of TCF1 targets.

The propensity of NELF chromatin binding and its action for TCF1 targets prompted us to discern a physical relationship between these two transcription factors. TCF1 protein levels were comparable in naïve *Nelfb$^{f/f}$* and KO CD8$^+$ T cells (Supplementary Fig. 6c), suggesting that NELF unlikely impacts TCF1 targets by regulating its expression. Using proximity labeling[45], we found that NELF was in close proximity with TCF1 (Fig. 6c). As a positive control, similar physical proximity was detected between NELFB and NELFE, another NELF subunit (Fig. 6c). This finding is consistent with the idea that NELF and TCF1 work at a common set of targets to regulate chromatin accessibility in naïve CD8$^+$ T cells.

Chromatin accessibility at transcriptionally regulatory regions often precedes transcription of associated genes[46,47]. We, therefore, conducted deep RNA-sequencing using *Nelfb$^{f/f}$* and KO CD8$^+$ T cells before and after in vitro TCR activation by anti-

CD3/CD28 plus IL2. While baseline transcriptomes were similar between *Nelfb$^{f/f}$* and KO cells, TCR-activated transcriptomes of WT and KO cells were quite distinct (Fig. 6d, Supplementary Fig. 6d, and Supplementary Data 1). Of note, gene set enrichment analysis (GSEA) showed that *Nelfb* deletion was associated with enriched gene signatures for TCF1 deficiency, T cell exhaustion, and aging signature (Fig. 6e). Furthermore, KO cells exhibited reduced gene signatures for memory T cells and fatty acid metabolism, a hallmark of memory T cells[48] (Fig. 6e). In keeping with our findings, GSEA analysis of published scRNA-seq data from human melanoma showed that high *NELFB* expression in tumor-infiltrating CD8$^+$ cells significantly correlated with gene signatures for memory T cell and fatty acid metabolism[49] (Fig. 6f). Thus, our data strongly suggest that NELF facilitates TCR-triggered transcription that favors memory T cell fate and mitigates T cell exhaustion and aging.

**NELFB overexpression boosts antitumor immunity**. To determine whether NELFB overexpression could enhance adaptive immunity, we established a T cell-specific transgenic mouse model (referred to as Tg hereafter, Supplementary Fig. 7a). In addition to elevated NELFB levels (Fig. 7a), the expression of NELFA and NELFC, two other NELF subunits, were also increased in CD8$^+$ T cells of Tg mice (Supplementary Fig. 7b), likely through stabilization of the entire NELF complex. In an in vivo competitive assay, KO or Tg CD45.2$^+$CD8$^+$ T cells were mixed at a 1:1 ratio with their corresponding control CD8$^+$ T cells carrying a congenic marker CD45.1$^+$, which were subsequently transferred into B16 tumor-bearing recipient mice. Tumor-infiltrating CD8$^+$ cells were analyzed 2–3 weeks post-transfer (Fig. 7b). As expected, KO cells were significantly

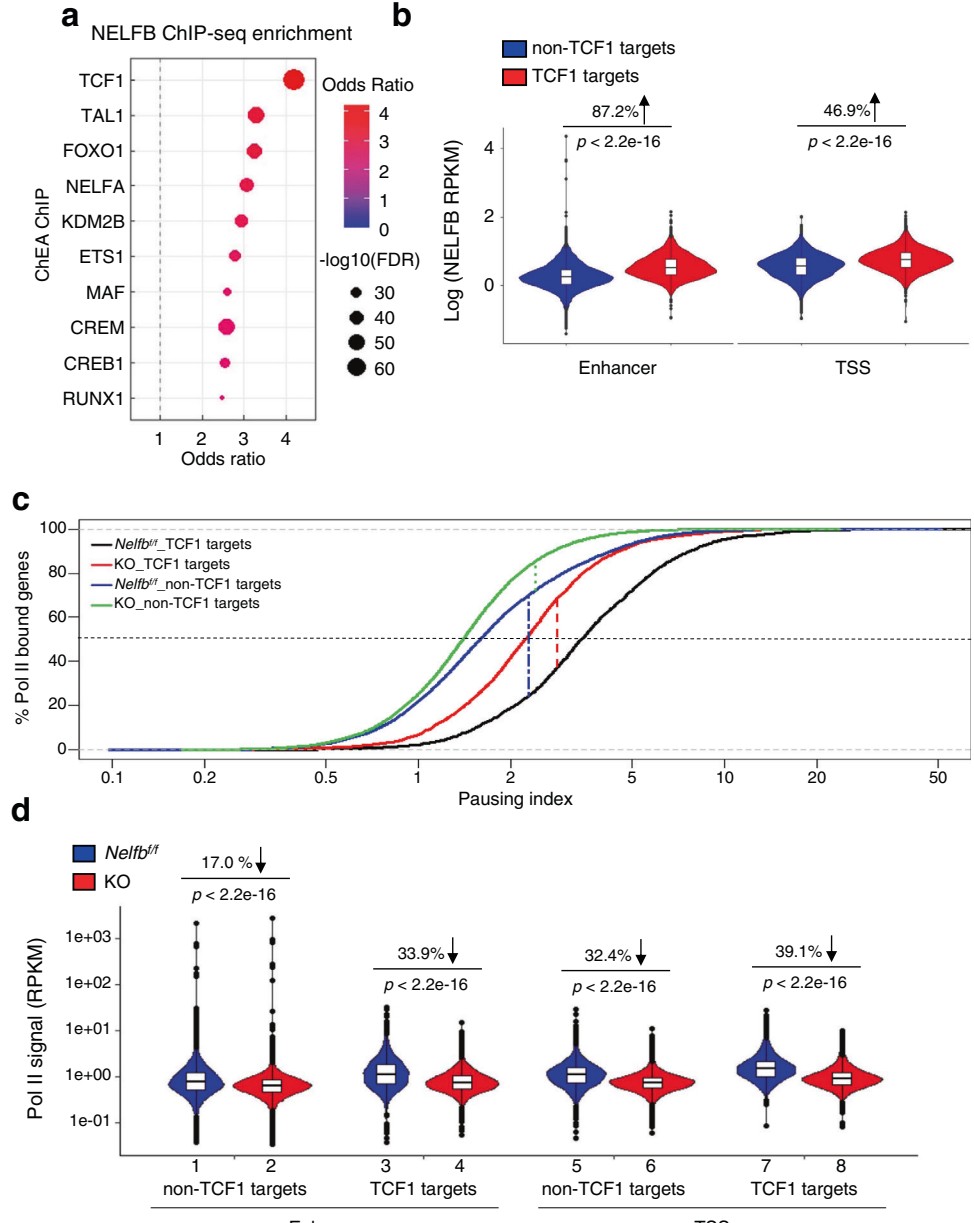

**Fig. 5 NELF-mediated Pol II pausing at *cis*-regulatory elements of TCF1 targets genes. a** Enrichment analysis of NELFB ChIP-seq of WT CD8+ T cells. **b** NELFB binding signals at enhancer and TSS regions of TCF1 and non-TCF1 targets of WT CD8+ T cells. **c** Cumulative curve of a pausing index for TCF1 and non-TCF1 targets in *Nelfb*^f/f and KO CD8+ T cells. Dashed vertical line denotes the maximum difference in vertical distance between two cumulative distributions. **d** Pol II signals at enhancer and TSS regions of TCF1 or non-TCF1 targets of *Nelfb*^f/f and KO CD8+ T cells. The bounds of the box represent the 25th and 75th percentiles of the interquartile range. The black line in the box interior represents the median of the data. The whiskers represent the minimum and maximum values of the data and the black dot outside the box and the whiskers represent an outlier. Violin plots were assessed by a two-sided Wilcoxon rank-sum test with continuity correction. For Pol II ChIP-seq analyses, $n = 2$ biological replicates/group.

outcompeted by control cells (Fig. 7c, d). In contrast, Tg cells comprised most of the tumor-infiltrating CD8+ cell population (Fig. 7e, f), indicating that NELFB-overexpressing T cells are superior to their WT counterparts. When purified WT and Tg CD8+ cells were transferred separately into *Rag1*^−/− immuno-deficient hosts followed by E0771 tumor challenge, Tg CD8+ T cells again exhibited more potent antitumor activity than their WT counterparts (Fig. 7g–i). Furthermore, mice receiving Tg CD8+ cells had significantly more total leukocytes (Fig. 7j) and CD8+ TILs (Fig. 7k). NELFB overexpression increased effector memory (Fig. 7l) and proliferative CD8+ T cell populations (Fig. 7m) while reducing the exhaustion marker PD1 expression

(Fig. 7n). In addition, NELFB overexpression increased both single and double IFNγ+/TNFα+ CD8+ cells (Fig. 7o–q). Col-lectively, our data suggest that NELF is a rate-limiting factor in boosting CD8+ T cell antitumor activity, which likely occurs through reducing T cell exhaustion and increasing memory and polyfunctionality.

T cell exhaustion and the lack of sustained persistence are the major barriers to successful CAR-T therapy[50]. We, therefore, sought to determine whether human NELFB (hNELFB) can boost T cell functionality in a more clinically relevant setting. We engineered a bi-cistronic CD19-specific CAR vector based on an established CAR construct, anti-CD19-28z[51] (anti-CD19-28z-

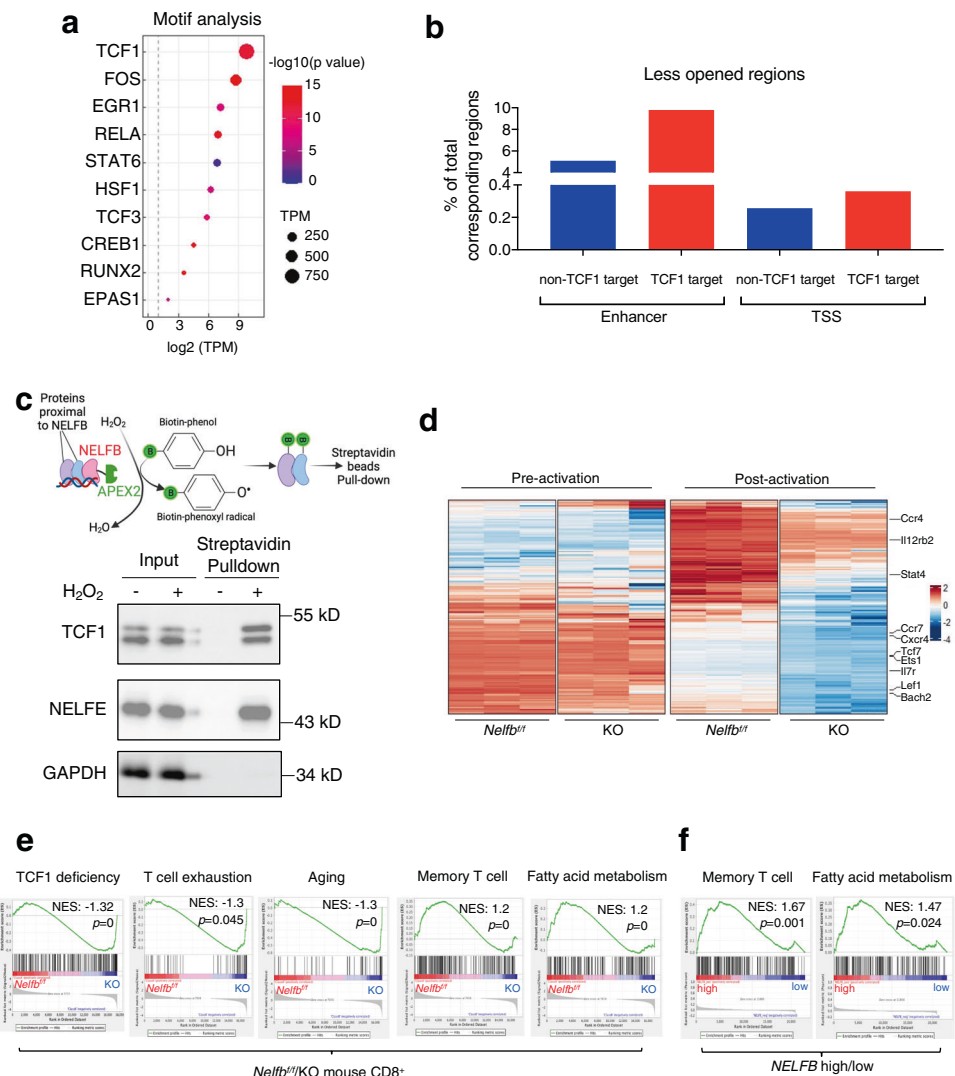

**Fig. 6 NELF is associated with TCF1 and facilitates chromatin accessibility and transcriptional program of TCF target genes. a** Motif analysis using less-opened chromatin regions in KO versus *Nelfb*[f/f] CD8[+] T cells; binomial test. **b** Less-opened regions in KO versus *Nelfb*[f/f] CD8[+] cells (% of total corresponding regions). For ATAC-seq analyses, $n = 2$ biological replicates/group. **c** Western blots for biotinylated protein by SDS-PAGE. NELFE and GAPDH served as positive and negative controls, respectively. The experiments were independently repeated three times with similar results. **d** Heatmaps showing genes with TCF1 binding and reduced ATAC-seq and RNA-seq signals in post-activated KO versus *Nelfb*[f/f] CD8[+] T cells. **e** KO CD8[+] cells are associated with signatures of TCF7 deficiency, T cell exhaustion, and aging while *Nelfb*[f/f] CD8[+] cells are associated with signatures of memory T cell and fatty acid metabolism. For RNA-seq analyses, $n = 3$ biological replicates/group. **f** *NELFB*[high] human melanoma TILs are associated with signatures of memory T cell and fatty acid metabolism. Empirical phenotype-based permutation tests were used for gene sets enrichment analysis. Source data are provided as a Source Data file.

P2A-hNELFB). Lentivirus-mediated transduction of primary human T cells resulted in overexpression of hNELFB (Supplementary Fig. 8a) without boosting CAR expression or altering the CD4:CD8 ratio during in vitro expansion (Supplementary Fig. 8b–d). In vitro expanded human T cells with hNELFB overexpression displayed an increased percentage of CD62L[+]CD45RA[+], a reported hallmark shared by naïve and memory stem cells[52], in both CD4[+] and CD8[+] populations (Fig. 8a, b). In a Raji lymphoma model using NSG immunodeficient mice, T cells carrying the parental anti-CD19-CAR-28z vector significantly prolonged the survival of tumor-bearing mice when compared with mice receiving PBS or mock-infected T cells (Fig. 8c). Of note, T cells with anti-CD19-CAR-28z-hNELFB conferred markedly superior host survival benefits over those with the parental CAR-T vector (Fig. 8c). In a solid tumor model, in which CD19 antigen was engineered in human breast cancer

cell line MDA-231, host mice receiving hNELFB-expressing CAR-T cells exhibited smaller tumor growth than those with parental CAR-T cells (Supplementary Fig. 8e). Furthermore, compared to parental CAR-T, hNELFB-expressing CAR-T conferred more tumor infiltration of CD8[+] and CD4[+] T cells (Supplementary Fig. 8f, g), and higher memory marker CD127 expression and fewer cells with the exhaustion markers TIM3[+]CD39[+] in both CD8[+] and CD4[+] populations (Supplementary Fig. 8h–k). Our data, therefore, provide the proof of principle that hNELFB overexpression could boost CAR-T anticancer immunotherapy.

## Discussion
In the current study, we defined a CD8[+] T cell-intrinsic function of the Pol II pausing factor NELF in regulating adaptive immunity. Upon genetic ablation of *Nelfb*, CD8[+] T cells exhibit

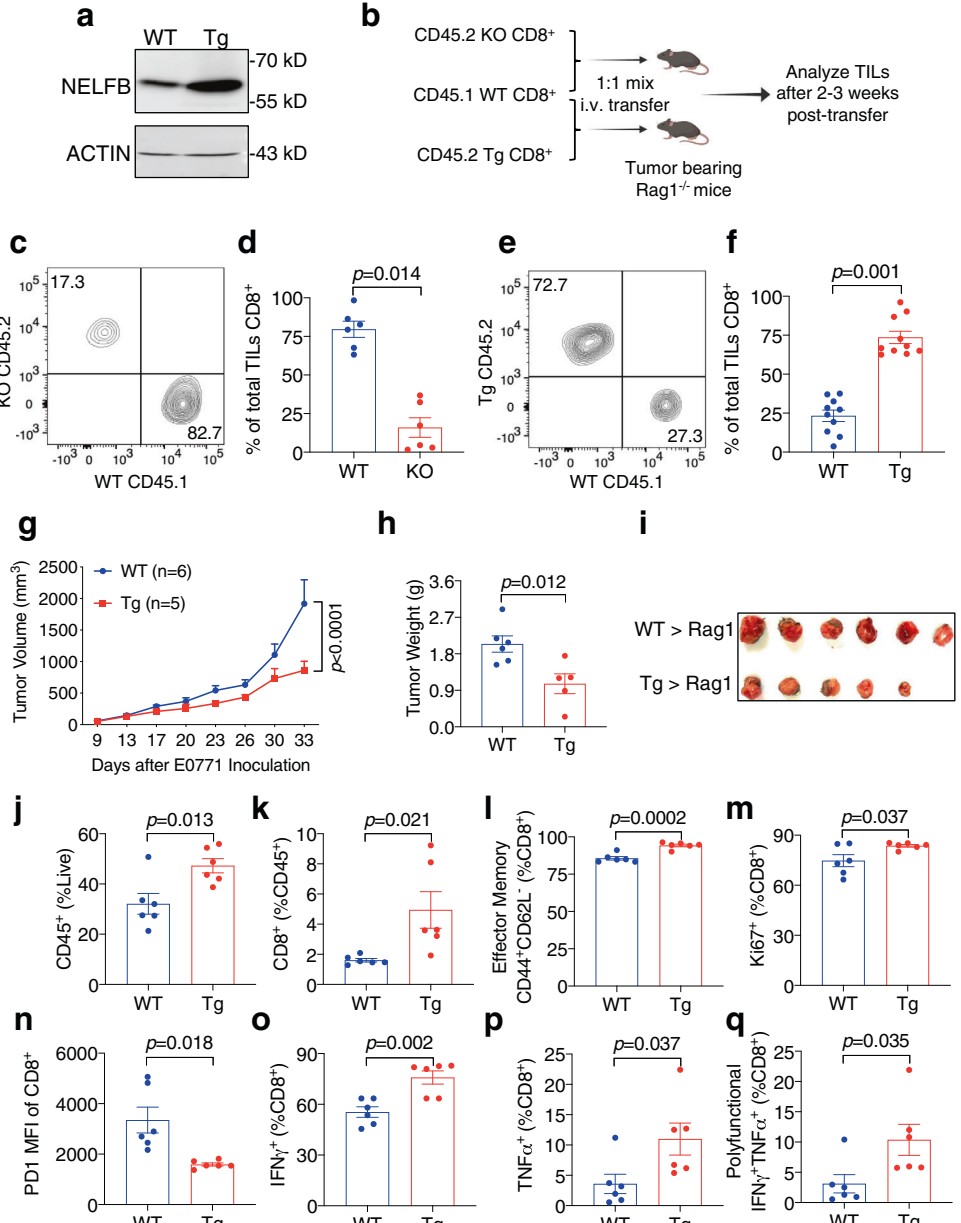

**Fig. 7 NELFB overexpression promotes the tumor-killing capacity of CD8$^+$ T cells. a** Representative Western blots for NELFB and ACTIN in WT/Tg splenic CD8$^+$ T cells. The experiments were independently repeated three times with similar results. **b** Scheme of competitive co-transfer assay. **c** Representative flow cytometry plot of TILs in the mice that received WT/KO CD8$^+$ T cells. **d** Percentage of total TILs CD8$^+$ cells in mice receiving WT/ KO CD8$^+$ T cells, $n = 6$/group. **e** Representative flow cytometry plot of TILs in mice receiving WT/Tg CD8$^+$ T cells. **f** Percentage of total TILs CD8$^+$ cells in mice receiving WT/Tg CD8$^+$ T cells, $n = 10$/group. **g–i** E0771 tumor growth curve (**g**), weight (**h**), and image (**i**) in $Rag1^{-/-}$ mice receiving WT or Tg CD8$^+$ T cells; WT ($n = 6$), Tg ($n = 5$). **j–q** TIL analysis for E0771 tumors in $Rag1^{-/-}$ mice receiving WT or Tg CD8$^+$ T cells, $n = 6$/group. **j** CD45$^+$ (% of total live cells), **k** CD8$^+$ (% of CD45$^+$), **l** effector memory CD44$^+$CD62L$^-$ (% of CD8$^+$), **m** Ki67$^+$ (% of CD8$^+$), **n** Median fluorescence intensity (MFI) of PD1, **o** IFNg$^+$ (% of CD8$^+$), **p** TNFa$^+$ (% of CD8$^+$), **q** polyfunctional IFNγ$^+$TNFα$^+$ (% of CD8$^+$). Data were presented as mean ± SEM; mean differences were compared using Student's *t*-test. Tumor curves were compared using two-way ANOVA followed by multiple comparisons. Two-sided tests were used. Source data are provided as a Source Data file.

accelerated apoptosis, impaired proliferation, loss of poly-functionality, and reduced memory cell populations, all of which likely contribute to defective antitumor immune responses. The effect of *Nelfb* KO on memory T cell response can be attributed to a direct role of NELF in the memory cell population or an indirect consequence of NELF functions in effector T cells, ana-logous to the findings from the genetic studies of other T cell transcription factors such as *Batf* and *Irf4*[53,54]. Combining scRNA-seq and differentiation trajectory analyses, we found that

T cell-specific deletion of *Nelfb* led to more terminally differ-entiated populations while diminishing naïve and memory pro-genitor pools. These findings are consistent with a recent study using a muscle stem cell (MuSC)-specific *Nelfb* KO model, in which *Nelfb* deletion impaired MuSC replenishment in response to repetitive muscle injury due to premature terminal differ-entiation of myogenic progenitors[28]. We point out that NELFB protein in CD8$^+$ T cells was incompletely depleted in our T cell-specific genetic model, similar to what was observed in other

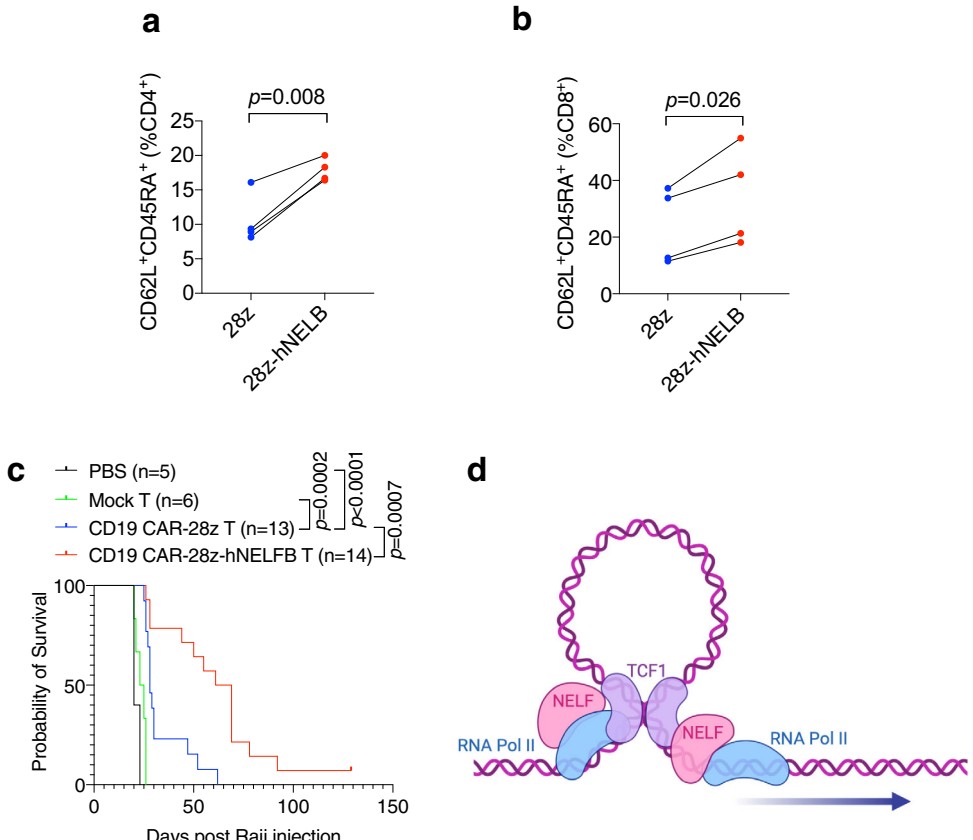

**Fig. 8 NELFB overexpression boosts human CAR-T efficacy. a, b** Percentages of CD62L$^+$CD45RA$^+$ cells within CD4$^+$ (**a**) and CD8$^+$ (**b**) populations of 28z or 28z-hNELFB CAR-T cells after 10 days of in vitro expansion, $n = 4$/group. **c** Survival curves of Raji tumor-bearing mice receiving PBS or mock T cells, CD19 CAR-28z T cells, or CD19 CAR-28z-hNELFB T cells. **d** Model of enhancer–promoter looping mediated by NELFB, Pol II, and TCF1. Mean differences were compared using Student's $t$-test. Log-rank (Mantel–Cox) tests were used for survival analyses. Source data are provided as a Source Data file.

tissue-specific *Nelfb* KO mouse models[26–28]. It is conceivable that both the protein amount of individual NELF subunits and the stability of the NELF complex can affect T cell lifespan and functionality. The robust phenotypes of our KO and transgenic mouse models provide compelling evidence for an important and rate-limiting role of NELF in sustaining CD8$^+$ cell functions in antitumor immunity.

APEX2-mediated proximal labeling confers cross-linking of neighboring proteins in the radius of 1–10 nm[55]. Therefore, our result strongly suggests physical proximity, but not necessarily direct protein–protein interaction, between NELFB and TCF1. Nevertheless, together with the genome-wide findings, the proximal labeling data are consistent with the notion that NELF and TCF1 are part of the same transcription complex co-occupied at a common set of transcriptional enhancers and promoters in CD8$^+$ T cells. It is also worth pointing out that our ATAC-seq was conducted in primary naïve CD8$^+$ T cells, whereas bulk RNA-seq was done in CD8$^+$ T cells before and after TCR activation. Unlike baseline transcriptomes prior to TCR activation, which were similar between *Nelfb*$^{f/f}$ and KO cells, TCR-activated transcriptomes of *Nelfb*$^{f/f}$ and KO cells were substantially divergent. Therefore, our data favor the model in which NELF-dependent chromatin openness prior to TCR activation precedes and likely facilitates TCR-stimulated transcription of TCF1 target genes (Fig. 8d).

NELF-dependent Pol II pausing at promoter-proximal regions is known to facilitate transcription by maintaining an accessible chromatin architecture[21] and stabilizing the transcription initiation complex[26]. In contrast, relatively little is known about the functional consequence of Pol II accumulation at transcriptional

enhancers. Our genome-wide survey of primary CD8$^+$ T cells supports the notion that NELF-dependent Pol II accumulation has even stronger effects on chromatin accessibility in enhancers versus promoters. While the exact mechanism by which enhancer-associated Pol II regulates transcription remains to be determined, we envision at least two plausible scenarios: (1) enhancer-associated Pol II may facilitate enhancer–promoter looping/interaction (Fig. 8d); and (2) Pol II-transcribed enhancer RNA could serve as the anchor for additional transcription factors/cofactors[44,56]. Histone-based epigenetic programming has been implicated in various stages of T cell development and differentiation[57]. For example, terminally exhausted and stem-like memory T cells have distinctive histone modification and chromatin accessibility profiles[58]. In addition, histone methyltransferase Suv39h1 has been shown to restrain the transcriptional program of memory T cell differentiation by enabling H3K9me3 deposition at memory-related genes[59]. We propose that NELF-mediated Pol II pausing may be part of the epigenetic programming that dictates the differentiation and function of memory T cells.

Our work suggests a strategy to improve outcomes of CD8$^+$ T cell-based adoptive cell therapy. NELFB overexpression in both human and mouse T cells significantly enhances antitumor immunity and prolongs the survival of tumor-bearing hosts. Consistent with our findings, TCF1 overexpression has been shown to boost antitumor immunity and thus proposed to enhance CAR-T efficacy[60]. We propose that NELF belongs to an expanding list of transcription factors whose overexpression bolsters CAR-T efficiency[52,61]. It will be of interest to determine whether these

transcription factors regulate the same or distinct transcriptional programs in T cell-based immunotherapy. The anti-CD19-28z CAR construct we used is more exhaustion-prone than newer generations of anti-CD19 CAR constructs carrying the 4-1BB or ICOS signaling domain[62]. To strengthen the translational potential of NELFB overexpression in boosting adoptive cell therapy, future experiments are needed to test NELFB overexpression in other CAR constructs with tonic signaling and exhaustion-prone phenotypes, such as GD2.28z CAR-T[63]. Because CD8[+] T cell exhaustion was observed as a pan-cancer phenotype[64], NELF function in mitigating immune exhaustion could be applied to multiple cancer types.

In summary, our current study shows that NELF in CD8[+] T cells works at both enhancers and promoters of TCF1 target genes to potentiate chromatin accessibility and transcriptional activation. We propose that the functional cooperation between NELF-dependent Pol II pausing and TCF1 is a mechanism that controls conversion between memory T cells and differentiated effector cells, ultimately dictating antitumor immunity and efficacy of cell-based immunotherapy against cancer.

## Methods

**Mice**. All animal protocols were approved by the Institutional Animal Care and Use Committee at George Washington University. *Nelfb*[f/f] and distal *Lck-cre* (*dLck-cre*) mice were created as previously described[24,65]. *Nelfb*[Tg] mice were generated by CRISPR-based gene editing (Cyagen Biosciences). Briefly, the gRNA to ROSA26, a donor vector containing CAG promoter-*loxP*-Stop-*loxP*-mouse *Nelfb* cDNA-polyA, and *Cas9* mRNA were co-injected into fertilized mouse eggs to generate targeted knockin offspring. F0 founder animals, identified by PCR followed by sequence analysis, were bred to WT mice and assessed for germline transmission and F1 mouse generation. T cell-specific *Nelfb* KO or transgenic mice were generated by crossing *dLck-cre* with *Nelfb*[f/f] or *Nelfb*[Tg], respectively. The Cre and floxed strain behaved the same as WT B6 mice; we, therefore, used the floxed mice as WT. *Rag1*[−/−] (stock no. 002216), NSG (stock no. 005557), and C57BL/6 CD45.1 congenic mice (stock no. 002014) were purchased from The Jackson Laboratory. Mice that are 10-week-old were considered adults and were used for tumor studies.

**In vivo tumor challenge and assessment**. For direct tumor challenge, syngeneic mammary tumor E0771 cells (5 × 10⁵ cells) (CH3 Biosystems, 940001) or AT-3 (2 × 10⁵ cells) (a generous gift from Dr. Scott Abrams at Roswell Park Comprehensive Cancer Center)[66] were inoculated into the fourth mammary fat pad of female C57BL/6 mice.

For adoptive CD8[+] T cell transfer experiments, total CD8[+] cells were isolated from mouse spleen using an EasySep™ Mouse CD8[+] T Cell Isolation Kit (STEMCELL Technologies, 19853). WT/KO (3.5 × 10⁶) or WT/Tg CD8[+] cells (5 × 10⁵) were transferred by tail vein injection into *Rag1*[−/−] recipients. E0771 (5 × 10⁵ cells) mammary tumor cells were inoculated into the fourth mammary fat pad one day after CD8[+] adoptive transfer.

Heat-inactivated tumor cell vaccination experiments were done using an established protocol with some modifications[35]. Briefly, 5 × 10⁶ heat-killed (boiled for 30 min) B16 mouse melanoma cells (ATCC, CRL-6475) were subcutaneously inoculated into the back flanks of mice. Two weeks after the initial vaccination, 5 × 10⁶ heat-killed B16 cells were subcutaneously inoculated into the back flanks of mice as a booster. Following another 2 weeks, 1 × 10⁵ live B16 cells were subcutaneously inoculated into the back flanks of mice and tumor growth was monitored.

OVA immunization was done following a previously published protocol[67]. Briefly, OVA emulsified in CFA (Hooke Laboratories, EK-0301) was subcutaneously injected at two sites on the backs of mice (0.05 ml per site). Two weeks later, OVA emulsified in IFA (Hooke Laboratories, EK-0311) was subcutaneously injected at one site on the backs of mice (0.1 ml) as a booster. Control CFA emulsions (Hooke Laboratories, CK-0301) and IFA emulsions (Hooke Laboratories, CK-0311) were used as the corresponding controls. After at least 2 weeks of booster injection, 2 × 10⁶ E.G7-OVA mouse T cell lymphoma cells (ATCC, CRL-2113) or 2.4 × 10⁶ B16-OVA mouse melanoma cells (generated by Dr. Tyler Curiel's lab at the University of Texas Health San Antonio) were subcutaneously injected into the backs of mice. All solid tumors were measured by digital calipers (tumor volume = 0.5 × length × width²). Survival analyses used animal death or tumor size >1000 mm³ as endpoints.

For CAR-T experiments, NSG mice were inoculated with Raji lymphoma cells (5 × 10⁵ in 100 μl PBS) through tail vein injections. One day after the tumor challenge, 1 × 10⁵ CAR-T cells in 100 μl PBS were administered through tail vein injections. Survival analyses used animal death or hind limb paralysis as endpoints. MDA-MB-231 human breast cancer cells were infected with lentiviruses expressing human CD19 (pLenti-GIII-EF1a, Abmgood, 154800610695). Cells were then selected by Puromycin, successful CD19 overexpression were confirmed by flow

cytometry using an anti-CD19 antibody (Biolegend, 302239). About 4 × 10⁵ MDA-MB231-hCD19 tumor cells were orthotopically injected into NSG mice. After 10 days, tumor sizes were measured and mice were randomized for receiving either mock treatment (PBS), 1.8 × 10⁶ 28z, or 1.8 × 10⁶ 28z-hNELFB CAR-T cells through tail vein injection. Tumor sizes were then monitored.

**Flow cytometry**. Tumor tissues were first minced and passed through 70 μm filters to obtain single-cell suspension. Cells were then stained with Viability Ghost Dye 510 (Tonbo Biosciences, 13-0870) for 10 min at 4 °C and blocked by anti-CD16/32 (Tonbo Bioscience, 70-0161) (1 to 100 dilution) at 4 °C for 10 min. Cells were further stained with anti-CD45 (Invitrogen, 11-0451-82), anti-CD3 (Tonbo Biosciences, 65-0031-U100), anti-CD8 (BD Pharmingen, 557654), anti-CD44 (BioLegend, 103041), anti-CD62L (BioLegend, 104424), anti-PD1 (BD Biosciences, 563059), anti-TIM3 (BioLegend, 119704), and anti-LAG3 (Invitrogen, 25223182) in staining buffer (2% FBS in PBS). For nuclear transcription factor staining, cells were permeabilized using a FoxP3/transcription factor staining kit (eBioscience, 00-5523-00) and then stained with anti-Ki67 (Invitrogen, 48-5698-82). For cytokine staining, cells were stimulated by anti-CD3/CD28 (Thermo Fisher, 11452D) at 37 °C overnight and then treated with BD GolgiPlug (BD Biosciences, 550583) at 37 °C for 5 h. After surface staining, cells were permeabilized using a BD Cytofix/Cytoperm kit (BD Biosciences, 554714) and stained with anti-IFNγ (BioLegend, 505826) and anti-TNFα (BioLegend, 506314). All antibodies were used at 1:150 dilution. Cells were then fixed with 1% paraformaldehyde and analyzed by BD FACSCelesta. Data were analyzed using BD FACSDiva and FlowJo software.

**In vitro CD8[+] T cell characterization**. For apoptosis detection, naïve CD8[+] T cells were isolated from mouse splenocytes using a negative selection protocol (STEMCELL Technologies, 19858). Cells were then activated by anti-CD3/CD28 (25 μL beads per million cells) (Thermo Fisher, 11452D) following the manufacturer's instructions. After 24 h of activation, cells were stained using an Annexin V Apoptosis Detection Kit (eBioscience, 88-8007) and analyzed by flow cytometry.

For CFSE labeling experiments, isolated naïve CD8[+] T cells were first labeled with a CFSE Cell Division Tracker Kit (BioLegend, 50-712-280) and activated by anti-CD3/CD28 (Thermo Fisher, 11452D) for 24 h. The activation beads were then removed by a magnet and recombinant mouse IL2 (R&D SYSTEMS, 402-ML-020) was added to the culture medium (10 ng/ml). Fresh IL2-containing medium was replaced every other day. Cell seeding density was maintained between 0.5–1 × 10⁶/ml. CFSE intensity was examined by flow cytometry.

To characterize cytokine production and exhaustion phenotypes, total CD8[+] T cells were first isolated from mouse splenocytes (STEMCELL Technologies, 19853). Cells were then activated by anti-CD3/CD28 (Thermo Fisher, 11452D) and subsequently cultured in an IL2-containing (R&D Systems, 402-ML-020) medium. Freshly prepared IL2-containing medium was replaced every other day. Anti-PD1 and anti-TIM3 were stained and examined by flow cytometry at the indicated timepoints. For cytokine production, cells were treated with BD GolgiPlug (BD Biosciences, 550583) at 37 °C for 5 h and permeabilized using a BD Cytofix/Cytoperm kit (BD Biosciences, 554714) and subsequently stained with anti-IFNγ and anti-TNFα.

**Bulk RNA-seq and single-cell RNA-seq analyses**. For bulk RNA-seq, total CD8[+] T cells directly isolated from 8-month-old mouse splenocytes were studied in the pre-activation stage. CD8[+] T activated by anti-CD3/CD28 and subsequently expanded in IL2-containing medium for 8 days were used as the post-activation stages. Total RNA was extracted by an RNeasy Mini Kit (Qiagen, 74104) following the manufacturer's instructions. RNA samples were processed further as follows. Briefly, about 500 ng total RNA was used for library preparation following the protocol for Illumina TruSeq stranded mRNA-seq. PolyA-containing mRNA was enriched and converted into first-strand cDNA by random primers and reverse transcriptase. Second-strand cDNA were then synthesized and final RNA-seq libraries were generated by PCR. An Illumina HiSeq 3000 platform was used to carry out 50 bp single-read sequencing.

For single-cell RNA-seq, total CD8[+] T cells were isolated from 10-month-old mouse spleens using a negative selection protocol according to the manufacturer's instructions (STEMCELL Technologies, 19853). Freshly isolated, >90% viable CD8[+] T cells were immediately loaded onto a Chromium Next GEM Chip G (10x Genomics, PN-1000127) and processed with the Chromium Controller (10x Genomics), with a target of 3000 cells per sample. Single-cell RNA-seq libraries were built using Chromium Next GEM Single Cell 3′ GEM, Library & Gel Bead Kit v3.1 (10x Genomics, PN-1000128), and barcoded with unique Illumina sample indexes (10x Genomics, PN-120262). Libraries were sequenced on an Illumina Hiseq instrument with a 10x Genomics-compatible configuration, and 50,000 reads per cell were targeted.

Sequencing data were demultiplexed and processed by Cell Ranger (version 3.1.0). STAR[68] was used to map reads to the mouse reference genome (refdata-cell ranger-mm10-3.0.0). The outputs of individual samples were loaded into the Seurat R package[69] (version 3.1.5). High-quality cells were filtered based on the number of genes detected (between 1000 and 5000) and the percentage of unique molecular identifiers (UMIs) mapped to mitochondrial genes (<12%). Individual samples were integrated, and principal components were calculated. The first ten principal

components were used for cell clustering and tSNE visualization. Data QC analysis for scRNA-seq were included in Supplementary Fig. 9a–d. Bimod test was used for marker gene expression comparison between different populations (Supplementary Fig. 4c, d).

**ChIP-seq and ATAC-seq assays**. For ChIP-seq assays, cells were first cross-linked using 1% formaldehyde for 10 min and subsequently terminated by 125 mM glycine at room temperature for 5 min. The cross-linking reagent was removed by spinning at 1000 g at 4 °C for 5 min. Cells were then washed with cold PBS three times. From this step until ChIP elution, all buffers were prepared with a freshly added cocktail of protease inhibitors (1 µg/ml leupeptin, 1 µg/ml aprotinin, 1 µg/ml pepstatin, and 1 mM phenylmethylsulfonyl fluoride). Cells were lysed on ice for 10 min using lysis buffer (5 mM HEPES, pH 7.9, 85 mM KCl, 0.5% Triton X-100). The supernatant was removed after spinning at $1600 \times g$ at 4 °C for 5 min, and pellets were resuspended with nuclei lysis buffer (50 mM Tris-HCl, pH 8.0, 10 mM EDTA, 1% SDS). Chromosomal DNA was sonicated using a probe sonicator on ice, then centrifuged at $14,000 \times g$ for 15 min, and the supernatant saved for immunoprecipitation. 10% of the sonicated DNA was saved as input. Antibodies used for ChIP include anti-Pol II (BioLegend; 664906) and anti-NELFB (Cell Signaling Technology, 14894 S). Sonicated DNA was incubated with antibodies at 4 °C overnight. Dynabeads Protein A (Thermo Fisher Scientific, 10002D) was added the following day and incubated for 2 h. After incubation, Dynabeads was washed as previously described[42]. Samples were subsequently eluted and reverse-cross-linked at 65 °C overnight. Immunoprecipitated chromatin and input chromatin were ethanol-precipitated and used for library construction.

Sequencing and bioinformatic analysis were conducted at the UT Health San Antonio Genome Sequencing Facility. Briefly, the size distribution of ChIP-DNA was checked by Fragment Analyzer High Sensitivity DNA assays (Agilent Technologies). 0.1–20 ng ChIP-DNA (100–400 bps) was used for ChIP-seq library preparation using SwiftBiosciences Accel-NGS 2 S Plus DNA Library Kit (SwiftBiosciences, 21024). ChIP-seq libraries were quantified by Qubit and Bioanalyzer, and then pooled for cBot amplification and sequenced with 50 bp single-read sequencing using an Illumina HiSeq 3000 platform. After sequencing, fastq files were generated with Bcl2fastq2. ChIP-seq data quality was checked by MultiQC (v1.9).

For ATAC-seq assays, freshly isolated naïve CD8$^+$ T cells from mouse spleen (STEMCELL Technologies, 19858) were washed and centrifuged at $500 \times g$ for 5 min. Cell pellets were then resuspended in a cryopreservation solution containing FBS and 10% DMSO. Approximately 100,000 frozen cells of each sample were used by the Active Motif to perform ATAC-seq and analyses. Briefly, cells were thawed at 37 °C. Cell nuclei were first isolated to perform Tn5 tagmentation and make libraries of open chromatin, as previously described[70,71]. Tagmented DNA was then purified using a MinElute PCR purification kit (Qiagen, 28004), amplified with ten cycles of PCR, and purified using Agencourt AMPure SPRI beads (Beckman Coulter). Afterward, paired-end (PE42) sequencing with a depth of 30 million reads (a total of 60 million reads) was performed using NextSeq 500 (Illumina). FRIP scores and peak counting were used as quality controls. Reads were aligned to the mouse genome (mm9) using the BWA algorithm. Duplicate reads were removed, and only uniquely mapped reads (mapping quality ≥1) and reads mapping as matched pairs were used for further analysis. Peaks were identified using the MACS 2.1.0 algorithm. Peak read numbers correlation were included in Supplementary Fig. 9e. For genomic analyses, active enhancers were defined by H3K27ac binding peaks at non-TSS regions in CD8$^+$ T cells, and active promoters/TSS were defined as H3K4me3 peaks that overlap with TSS in CD8$^+$ T cells[72]. TCF1 targets were defined using previously published CD8$^+$ TCF1 ChIP-seq data[41].

**Co-transfer experiment**. Splenic CD8$^+$ T cells were isolated from congenic marked WT C57BL/6 (carrying CD45.1) or *dLck-cre;Nelfb*$^{f/f}$ mice (carrying CD45.2), mixed at a ratio of 50:50, and intravenously transferred into B16 melanoma tumor-bearing *Rag1*$^{-/-}$ immunodeficient mice. When comparing WT versus *Nelfb*-overexpressing cells, WT (CD45.1) or *dLck-cre;Nelfb*$^{Tg}$ CD8$^+$ (CD45.2) were co-transferred. A total of $2 \times 10^4$ cells were transferred in 100 µl PBS injected via the tail vein. Percentages of CD45.1 (BioLegend, 110706) and CD45.2 (BioLegend, 109814) in tumor-infiltrating CD8$^+$ cells were assessed by flow cytometry after 2–3 weeks post-transfer.

**Proximity labeling and western blotting**. APEX2-mediated proximity biotinylation was done as previously described[73]. Briefly, APEX2-encoding DNA sequences were fused to the mouse *Nelfb* gene using standard molecular cloning techniques, and the overexpression plasmid was packaged with helper plasmids in 293 T cells to generate lentivirus stocks. Afterward, Jurkat cell lines (ATCC, TIB-152) were infected with the lentivirus and a stable cell line was selected using neomycin. Biotin-phenol labeling was conducted with 30 min incubation in 500 µM biotin-phenol. Cells were then exposed to 1 mM H$_2$O$_2$ at room temperature for 1 min. The reactions were then stopped with ice-cold Dulbecco's phosphate-buffered saline with quenchers (10 mM sodium azide, 10 mM sodium ascorbate, and 5 mM Trolox). Cells were then pelleted by centrifugation and lysed by RIPA lysis buffer containing 1 mM PMSF, 5 mM Trolox, 10 mM sodium ascorbate, and 10 mM sodium azide. A slurry of streptavidin magnetic beads (NEB, S1420S) was incubated with cell lysate and rotated at room temperature for 1 h. The beads were subsequently washed and boiled to elute biotinylated proteins. Cells without H$_2$O$_2$ exposure served as negative controls. The resultant protein lysates were analyzed using standard Western blotting techniques. Primary antibodies included anti-TCF1 (CST, 2203 S), anti-NELFE (Proteintech, 10705-1-AP), and anti-GAPDH (Bio-Rad, 12004167). Other antibodies used for Western blotting included anti-NELFB (Cell Signaling Technology, 14894 S), anti-NELFA (Proteintech, 10456-1-AP), and anti-NELFC (Cell Signaling Technology, 12265 S). All antibodies were used at 1:1000 dilution.

**CAR-T generation and characterization**. A lentiviral vector pELPS-CAR19-28z was a generous gift from Dr. Carl H. June's lab at the University of Pennsylvania[51]. The human *NELFB* gene coding sequence was inserted downstream of the CAR19-28z-encoding sequence, with the P2A cleavage sequence in between (Gene Universal, Inc.). Ultra-purified high-titer viruses for both unmodified and modified lentiviral vectors were packaged and generated by VectorBuilder Inc.

Freshly isolated (STEMCELL Technologies, 200-0046) or frozen (STEMCELL Technologies, 70024) human peripheral blood pan-T cells were stimulated by Dynabeads™ Human T-Activator CD3/CD28 (Gibco, 11131D) following the manufacturer's instructions. After 24 h of activation, high-titer CAR virus, recombinant human IL2 protein (Gibco, PHC0027), and polybrene (MilliporeSigma, TR1003G) were added to the wells and spinoculated at $800 \times g$ at 32 °C for 90 min. Infected T cells were expanded in vitro for 10 days in 10 ng/ml human recombinant IL2 (Gibco, PHC0027) containing RPMI (Corning, 45000-412) media before being injected into NSG mice via the tail vein. Subsets of cells were expanded in vitro, stained by anti-CD4 (BioLegend, 317418), anti-CD8 (BioLegend, 344710), anti-CD45RA (BioLegend, 304112), and anti-CD62L (BioLegend, 304806), and analyzed by flow cytometry using BD FACSCelesta. In vitro expanded cells were stained by FITC-labeled human CD19 (ACROBiosystems, CD9HF2H225UG) and subsequently purified by Anti-FITC MicroBeads (Miltenyi Biotec, 130-048-701) before the detection of NELFB overexpression by western blot or CAR expression by FITC-labeled human CD19. All antibodies were used at 1:100 dilution.

**Statistics**. Mean differences between the two groups were tested using Student's *t*-test. Mean differences between three or more groups were tested using one-way ANOVA. Tumor curves were compared using two-way ANOVA followed by multiple comparisons. Two-sided tests were used. Survival analyses were done by Log-rank (Mantel–Cox) and Gehan–Breslow–Wilcoxon tests. GSEA was done using GSEA software[74]. To analyze human melanoma tumor-infiltrating lymphocytes (TIL), we downloaded the single-cell RNA-seq dataset from GSE72056[49]. Activated CD8$^+$ TILs (CD8a ≥5 and CD44 ≥2) were used for GSEA as previously described[75]. Population distributions in scRNA-seq data analyses were examined using Chi-square tests. Wilcoxon rank-sum test with continuity correction was used for genomic analysis. Statistics were performed using GraphPad Prism software. $p < 0.05$ was considered significant.

**Reporting summary**. Further information on research design is available in the Nature Research Reporting Summary linked to this article.

## Data availability
The sequencing data that support the findings of this study have been deposited to NCBI Gene Expression Omnibus under accession codes GSE182862. Human melanoma tumor-infiltrating lymphocytes (TIL) data were downloaded from GSE72056[49]. The remaining data were available within the Article, Supplementary Information, or Source Data file. Source data are provided with this paper.

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

## Acknowledgements

We thank Drs. Tyler Curiel, Carl June, and Scott Abrams for the B16-OVA cell line, anti-CD19 CAR plasmids, and AT-3 cells, respectively. We also thank Dr. Xiujie Sun and Ms. Madeleine Prevost for technical assistance with tail vein injection and mouse genotyping, respectively. The model demonstration was created with BioRender.com. The work was supported by grants to R.L. from NIH (CA246707 and CA220578) and the Keck Foundation, to Y.H. from NIH (CA212674) and the Congressionally Directed Medical Research Program (W81XWH-17-1-0008), to C.N.Y. from NIH (R01HL141393 and R01DK117007). The Genome Sequencing Facility at UT Health San Antonio is supported by NIH-NCI P30 CA054174 (Mays Cancer Center at UT Health San Antonio), an NIH Shared Instrument grant 1S10OD021805-01 (S10 grant), and a CPRIT Core Facility Award (RP160732).

## Author contributions

R.L. managed and oversaw the overall project. R.L. and B.W. designed the experiments and wrote the manuscript. B.W., X.Z., H-C.C., H.P., B.Y., P.M., L.Q., and H.S. carried out the experiments. B.W., X.Z., C.N.Y., E.Y., Y.H., N.Z., and R.L. analyzed the data.

## Competing interests

R.L. and B.W. are co-inventors of a pending patent (application number 62/982,514) filed by The George Washington University on the therapeutic utility of NELFB in CAR-T therapy. The remaining authors declare no competing interests.
