## [Peer Review File · Nature Communications]

Reviewers' Comments:

Reviewer #1:

Remarks to the Author:

Wu et al show that perturbation of the NELF complex by adult genetic ablation of NELFB specifically in CD8+ T cells in a transgenic mouse model leads to CD8+ T cell-appropriate defects in vivo. The model for adult cell type specific genetic knockout of NELFB is specific and powerful. All subunits of the NELF complex are reduced in levels upon NELFB ablation only in CD8+ cells, which is shown convincingly. One of the more powerful experiments is overexpression of NELFB, which leads to restoration of the defect caused by depletion above the wild type baseline. This study shows that cells are acutely sensitive to the levels of NELF in terms of their physiological responses.

Questions:

The number of defects in mutant mice are shown in the beginning of the manuscript. It is clear from the transfer experiment that the defects are confined to CD8+ cells. However, how independent are these defects? The assays shown in the first and second chapters of results are indeed different, but is the defect in T-cell mediated recall different from defective antitumor response? Are these different experiments showing the same thing or different things mechanistically?

A related question is about the properties of mutant cells. The mutant model is normal in terms of precursors. Are mature cells being continuously produced in these mutant mice? If so, is it possible that the life span and the function of these cells is limited by the amount and stability of protein NELF remaining after the gene is disrupted? In other words, the functional defects of these mutant cells are not due to absence of NELF, but a decrease in NELF levels. Maybe mention in discussion?

Pausing index is a blunt instrument. Are there example genes with affected pausing (browser tracks) that can be included in the main figure to illustrate changes in PI? A sigmoidal curve encompassing all genes is not as informative.

Regarding NELF facilitating chromatin accessibility. Has NELF function or presence on DNA been shown in the absence of Pol II? NELF seems to accompany initiating Pol II everywhere so it is not clear if a statement can be made that NELF plays a role in chromatin accessibility. This statement implies that NELF would play that role alone or would be a "pioneer player". A counterpoint is that ablation of any subunit of Pol II or a GTF might lead to similar effects on chromatin accessibility. It is inherently hard to separate the effects of individual components of Pol II machinery, including NELF, on chromatin accessibility based on depletion, so maybe there is a way to tone down this conclusion?

In relation to RNA-seq experiments. Are there lists of differentially expressed genes pre and post activation pertinent to figure 6 between each condition, particularly between post-activation purple and dark green cells (activated WT and activated mutant)? A PCA plot may not be sufficient to ascertain differences in WT and KO activated cells.

There are typos in the text. The first paragraph of introduction has several (antigen should be antigens, for example, etc).

Reviewer #2:

Remarks to the Author:

The authors have addressed previous questions and comments, and have added new data that strengthen their claims. I now support publication of this manuscript. My only remaining suggestion is that the reviewer figure showing data for a panel of T cell associated TFs be included as part of the extended data - in my view, it only adds to the surprising result on TCF-associated NELF activity in enhancers and provides some important context.

Reviewer #3:

Remarks to the Author:

I have gone through the extensive changes made in response to my and the other reviewers' comments. The additional experimental data and increased clarity of text, in my view make the m/s now suitable for publication

RESPONSE TO REFEREES' COMMENTS

Notes to the Referees:

#1: Referees' comments in their entirety are copied below in *italics*. Point-by-point responses are shown in **blue**.

#2: New table in the revised manuscript are cited in **red** in our responses below.

Reviewer #1 (Remarks to the Author):

Wu et al show that perturbation of the NELF complex by adult genetic ablation of NELFB specifically in CD8+ T cells in a transgenic mouse model leads to CD8+ T cell-appropriate defects in vivo. The model for adult cell type specific genetic knockout of NELF B is specific and powerful. All subunits of the NELF complex are reduced in levels upon NELFB ablation only in CD8+ cells, which is shown convincingly. One of the more powerful experiments is overexpression of NELFB, which leads to restoration of the defect caused by depletion above the wild type baseline. This study shows that cells are acutely sensitive to the levels of NELF in terms of their physiological responses.

Questions:

The number of defects in mutant mice are shown in the beginning of the manuscript. It is clear from the transfer experiment that the defects are confined to CD8+ cells. However, how independent are these defects? The assays shown in the first and second chapters of results are indeed different, but is the defect in T-cell mediated recall different from defective antitumor response? Are these different experiments showing the same thing or different things mechanistically?

The analysis from tumor-free (Supplementary Fig. 1) and tumor-challenged mice (Fig. 1) suggests a defective memory T cell phenotype in KO mice, a notion further corroborated by findings from the tumor antigen-vaccinated memory recall experiment. It is well established that optimal primary antitumor response and T-cell mediated recall rely on both effector fate differentiation and persistent memory T cells [PMID: 30928016]. Therefore, it is highly likely that the *Nelfb* KO effect manifested in these two experiments are mechanistically related. Future work is needed to dissect the specific differentiation stage(s) (effector vs memory) that is defective in KO T cells by using immunologically more defined models such as virus infection and inducible KO mouse models.

A related question is about the properties of mutant cells. The mutant model is normal in terms of precursors. Are mature cells being continuously produced in these mutant mice? If so, is it possible that the life span and the function of these cells is limited by the amount and stability of protein NELF remaining after the gene is disrupted? In other words, the functional defects of these mutant cells are not due to absence of NELF, but a decrease in NELF levels. Maybe mention in discussion?

We concur with the reviewer that naïve KO T cells are most likely capable of giving rise to effector T cells. It is the latter that has a higher propensity for exhaustion and senescence. Given that the Cre action may not lead to simultaneous deletion of both flox alleles in all *Nelfb*^{f/f} T cells, the observed KO phenotype could be partly due to reduced amount, rather than complete absence of, NELF. We now discuss this possibility in the manuscript.

Pausing index is a blunt instrument. Are there example genes with affected pausing (browser tracks) that can be included in the main figure to illustrate changes in PI? A sigmoidal curve encompassing all genes is not as informative.

The browser tracks for one representative gene, *Ccr7*, are included in the Supplementary Figure 6b.

Regarding NELF facilitating chromatin accessibility. Has NELF function or presence on DNA been shown in the absence of Pol II? NELF seems to accompany initiating Pol II everywhere so it is not clear if a statement can be made that NELF plays a role in chromatin accessibility. This statement implies that NELF would play that role alone or would be a “pioneer player”. A counterpoint is that ablation of any subunit of Pol II or a GTF might lead to similar effects on chromatin accessibility. It is inherently hard to separate the effects of individual components of Pol II machinery, including NELF, on chromatin accessibility based on depletion, so maybe there is a way to tone down this conclusion?

Our CHIP-seq and ATAC-seq results showed that NELF KO reduces Pol II accumulation and chromatin accessibility preferentially at TCF1-targeting regions. While we are *not* proposing that NELF is a stand-alone chromatin remodeler, our data are consistent with the notion that NELF-enforced Pol II accumulation at the enhancers/promoters of TCF1 targets helps keep the local chromatin in an open state. But we agree with the reviewer that further work would be needed to determine whether this proposed function of NELF/Pol II can be distinguished from their chromatin-independent activities in transcription initiation and elongation.

In relation to RNA-seq experiments. Are there lists of differentially expressed genes pre and post activation pertinent to figure 6 between each condition, particularly between post-activation purple and dark green cells (activated WT and activated mutant)? A PCA plot may not be sufficient to ascertain differences in WT and KO activated cells.

Supplementary Table 1 includes all differentially expressed genes pre and post activation.

There are typos in the text. The first paragraph of introduction has several (antigen should be antigens, for example, etc).

Corrected.

Reviewer #2 (Remarks to the Author):

The authors have addressed previous questions and comments, and have added new data that strengthen their claims. I now support publication of this manuscript. My only remaining suggestion is that the reviewer figure showing data for a panel of T cell associated TFs be included as part of the extended data - in my view, it only adds to the surprising result on TCF-associated NELF activity in enhancers and provides some important context.

The graph for Pol II signals at various TFs and locations is included as the Supplementary Figure 5b.

Reviewer #3 (Remarks to the Author):

I have gone through the extensive changes made in response to my and the other reviewers' comments. The additional experimental data and increased clarity of text, in my view make the m/s now suitable for publication.

We thank the reviewer for his/her constructive and insightful suggestions.